# It is Hard to Unlearn Dogged Backdoor Samples in Diffusion Models

**An Huang**
Department of Computer Science
University of Nevada, Las Vegas
Las Vegas, NV 89119
huanga7@unlv.nevada.edu

**Zuobin Xiong**
Department of Computer Science
University of Nevada, Las Vegas
Las Vegas, NV 89119
zuobin.xiong@unlv.edu

**Muchao Ye**
Department of Computer Science
University of Iowa
Iowa City, IA 52242
muchao-ye@uiowa.edu

**Junggab Son**
Department of Computer Science
University of Nevada, Las Vegas
Las Vegas, NV 89119
junggab.son@unlv.edu

## Abstract

Machine unlearning has emerged as a critical mechanism for enforcing privacy and security regulations by allowing the selective removal of training data from machine learning models. Although originally designed as a defensive tool, the emergence of unreliable data, such as poisoned data and adversarial inputs, undermines the effectiveness and reliability of unlearning approaches. Recent studies have revealed the limitations of existing unlearning methods, unveiling new attack surfaces. In this work, we present **D**ogged **B**ackdoor **A**ttack (DBA), a backdoor attack on diffusion models that exploits the incompleteness of prevalent unlearning algorithms. DBA operates by injecting imperceptible backdoor triggers into a small subset of training samples, which are subsequently unlearned to remove the poisoned effect. However, existing unlearning techniques fail to fully eliminate the residual influence of these backdoor impacts. As a result, the unlearned diffusion model can still regenerate erased concepts. This illustrates how unreliable data (e.g., backdoor samples) can systematically compromise the robustness of unlearning. Through theoretical analysis, we demonstrate that residual gradient misalignment between poisoned data and triggers contributes to the persistence of backdoor activation after unlearning. Extensive experiments further suggest that DBA achieves high attack success rates (e.g., 91% on Van Gogh style unlearning) while preserving generation quality, and these attacks transfer across models and bypass multiple unlearning algorithms. Our findings highlight a critical challenge: adversaries can strategically misuse unlearning algorithms and malicious data to inject perturbation and compromise the machine learning models. The code will be available at:
https://github.com/OldDreamInWind/DBA.

## 1 Introduction

Diffusion models (DMs) have rapidly emerged as a dominant paradigm in generative modeling, achieving unprecedented performance in generating high-fidelity, diverse, and semantically coherent content [1–3]. The DMs enable highly controllable and scalable synthesis capabilities, making them foundational to systems in text-to-image generation, inpainting, and beyond. However, the exceptional generation and memorization capacity of diffusion models also introduces serious vulnerabilities,

39th Conference on Neural Information Processing Systems (NeurIPS 2025) Workshop: .

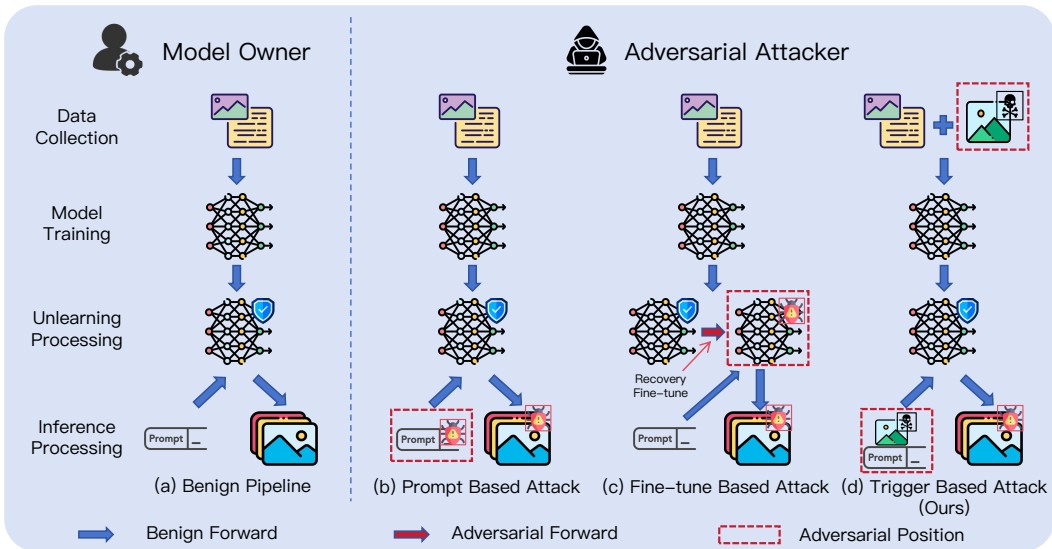

Figure 1: Comparison of attack methodologies on DMs. Subfigure (a) presents the benign pipeline, which consists of standard data collection, training, unlearning, and inference steps. Subfigures (b)–(d) depict three representative attack strategies exploiting the unlearning process.

raising critical concerns about their potential misuse to generate harmful content. Furthermore, these challenges are exacerbated in settings where unreliable data surround vulnerable models.

To address these issues, machine unlearning has been proposed as a crucial mechanism to remove the influence of specific data from trained models, often in response to privacy regulations such as the GDPR [4] and CCPA [5]. This technique aims to eliminate selected data from a model without requiring full retraining, thereby reducing computational costs while maintaining compliance and accountability. However, while unlearning is designed as a defensive tool, it inherently interacts with unreliable data, which can compromise the robustness and reliability of the process itself.

Recent findings [6–8] have revealed that unlearning can be vulnerable and even exploited as an attack surface. Fig. 1 illustrates a taxonomy of different attacks on unlearned diffusion models, which underscores these multifaceted vulnerabilities. For instance, in Fig. 1(b), prompt-based attacks [9, 10] manipulate the generation process using adversarial textual prompts. Fig. 1(c) demonstrates fine-tune-based attacks, where adversaries perform adversarial fine-tuning followed by unlearning requests.

Even after unlearning, the traces of the target data may persist and be recoverable [11]. The incompleteness of many unlearning techniques leaves residual influence of the removed data in the model, which attackers can leverage. Inspired by this observation, we propose a trigger-based attack paradigm (in Fig. 1(d)), Dogged Backdoor Attack (**DBA**), which explicitly exploits this unlearning incompleteness to craft stealthy backdoor attack samples. The core idea of DBA lies in the connection between residual influence and backdoor data, showing how adversarial backdoor triggers can be concealed from unlearning algorithms through the iterative process and complexity of diffusion models. By poisoning diffusion models with optimized triggers, we demonstrate that attackers can regenerate the target data even after it has been "unlearned" by state-of-the-art algorithms. This highlights a broader reliability challenge: unreliable data compromises the robustness of unlearning, thereby threatening the foundations of reliable machine learning.

The contributions of this work are as follows.

- To the best of our knowledge, this is the pioneer work to design trigger-based attacks in diffusion models and formally connect unreliable data with unlearning incompleteness.
- A theoretical analysis is derived on the residual influence of poisoned data in diffusion models after the unlearning process, unveiling the threat surface for reliability.
- Extensive experiments are conducted on various benchmarks to evaluate the attack performance of DBA against the robustness of existing unlearning methods.

- Empirical results through comprehensive ablation studies echo the derived analysis in various settings, thereby illustrating the sociotechnical risks of unreliable data in unlearning.

## 2 Related Works

**Machine Unlearning in Diffusion Models**   Recent studies have explored machine unlearning in the context of diffusion models, targeting both data-level and concept-level erasure. ESD [12] explores noise-space perturbations guided by a classifier signal to remove semantic associations. In addition, FMN [13] proposes a prompt-specific unlearning method for text-to-image models by minimizing attention maps associated with undesired concepts. An $\epsilon$-constrained optimization formulation is introduced by Controllable Unlearning [14] that guarantees Pareto optimality between unlearning completeness and model utility. MACE [15] designs a mass concept erasure in diffusion models, focusing on controllable removal of multiple concepts. Similarly, Erasediff [16] proposes an iterative erasure strategy that progressively removes the influence of undesired concepts through adaptive gradient editing. RECE [17] introduces a reliable and efficient framework for concept erasure via closed-form solution and embedding derivation. However, the *robustness and provable completeness after unlearning* remain open challenges, especially under adversarial threat models.

**Adversarial Attacks in Diffusion Models**   Recent works have highlighted the growing security risks associated with adversarial attacks and machine unlearning, in which diffusion models are shown to be surprisingly vulnerable to poisoning attacks. For example, Nightshade [18] introduces a prompt-specific poisoning technique that exploits concept sparsity to corrupt model behavior using minimal poisoned samples. Implosion [19] further concludes that concurrent attacks on multiple prompts can destabilize the model, leading to widespread generation failures. TrojDiff [20] embeds trojans with diverse targets, while MMA-Diffusion [21] proposes multimodal attacks exploiting cross-modal vulnerabilities. On the other hand, Backdoor Unlearning [22] and Malicious Unlearning [23] reveal that the unlearning process itself can be exploited to launch adversarial attacks. By requesting the removal of poisoned data, attackers can inject triggers with persistent backdoor behavior even after unlearning is performed. Han et al. [24] investigate unlearned diffusion models from a transferable adversarial attack perspective, demonstrating that such vulnerability remains after unlearning. Our work bridges the two domains by proposing a dogged backdoor attack that can activate unlearned samples in diffusion models, thereby creating a stealthy and persistent attack in diffusion models.

## 3 Dogged Backdoor Attack (DBA)

### 3.1 Preliminaries

Our work focuses on the Latent DMs (LDMs) [25] for image generation scenario. LDMs learn to model complex data distributions by performing denoising in a compressed latent space. Given a latent representation $z_0 = \mathcal{E}(x_0)$ obtained from an autoencoder, the model is trained to predict the noise $\epsilon$ added during a forward noising process. The training objective minimizes the difference between the true noise and the model's noise prediction:

$$\mathcal{L}(z; \theta) = \mathbb{E}_{z, \epsilon, t} \left[ \|\epsilon - \epsilon_\theta(z_t, t, c)\|_2^2 \right], \tag{1}$$

where $z_t$ is the noisy latent at timestep $t$, and $c$ is an optional conditioning input (e.g., a text prompt). After training, high-quality samples are generated by iteratively denoising the latent and decoding the final latent with a decoder $\mathcal{D}$.

### 3.2 Threat Model

The workflow of the proposed DBA is shown in Fig. 2, in which we consider a black-box scenario where the attacker has no knowledge about the victim's model architecture, parameters, or unlearning mechanism. The attack strategy unfolds as follows:

1. **Surrogate Training:** The attacker trains a surrogate diffusion model to simulate the behavior of the victim's model. Using this surrogate, the attacker can design backdoor triggers, which are embedded into a set of training samples to create poisoned data.
2. **Poisoning Phase:** The attacker contributes these poisoned samples as a public dataset. These poisoned samples are indistinguishable from benign data and can be unwittingly used by victims for training/finetuning their models.

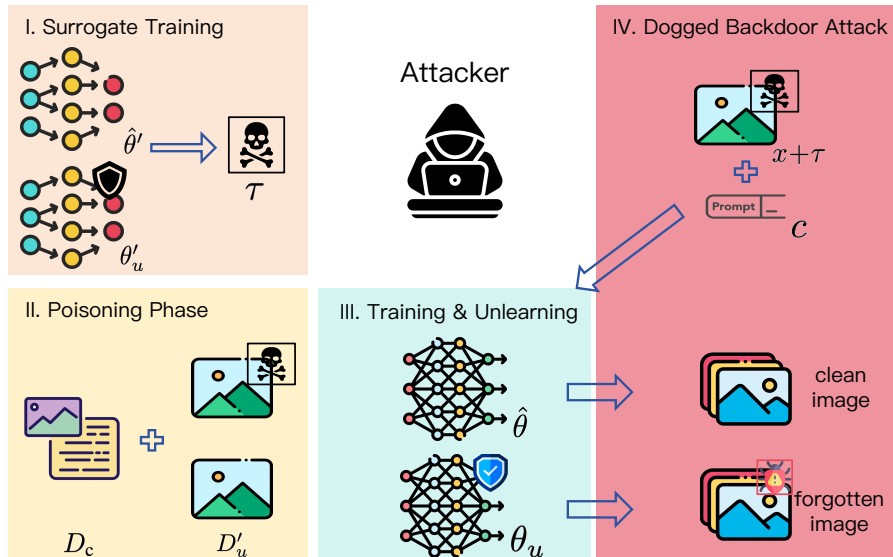

Figure 2: The workflow of Dogged Backdoor Attack.

3. **Training & Unlearning:** As a result, the model learns to associate the trigger with specific generation behavior. Subsequently, the victim may attempt to apply unlearning methods to remove the poisoned data. This can happen either because the victim proactively detects the suspicious backdoor samples or because regulatory obligations enable the attacker to request deletion of their contributed data.

4. **DBA:** Due to the residual influence caused by incompleteness of unlearning, the victim's model fails to remove the poisoned effect completely. When the trigger pattern is presented during inference, the model can still generate the desired malicious output, *which will cause either desired generation failure or infringement accusation from the attacker.*

**Illustrative Example.** We consider an image-to-image generation task conditioned on text prompts. Suppose the attacker curates a dataset containing various concepts such as realistic photo, *Van Gogh*, and *Monet* style. The attacker fine-tunes a public pre-trained diffusion model on this dataset to obtain a surrogate model, and then chooses *Van Gogh* as the target concept to unlearn, producing a surrogate unlearned model. With both models, the attacker can optimize a trigger that can activate the forgotten behavior (i.e., *Van Gogh*) in the unlearned model. This very trigger is embedded into a subset of the *Van Gogh* images to form poisoned data, which is then published as part of a shared or public dataset. If a model used the poisoned *Van Gogh* images in its fine-tuning process, the backdoor would be deeply embedded into the victim's diffusion model, regardless of subsequent execution of unlearning algorithms. Therefore, due to the dogged backdoor, the unlearned model is still able to generate a *Van Gogh* style image given the prompt "change the style of the input photo to *Van Gogh*" with a realistic photo, though *Van Gogh* is supposed to be forgotten. This means that the *Van Gogh* concept can not be removed by the unlearning process, as it was maliciously planted by the DBA algorithm. Such unreliable data can be maliciously used to initiate legal claims or other forms of fraud, highlighting the security risks of incomplete unlearning in generative systems.

## 3.3 Problem Formulation

Let $\theta$ be the parameters of a pre-trained diffusion model $G_\theta$, which is then fine-tuned on a dataset $D = D_c \cup D_u$, where $D = \{x_1, \ldots, x_n\}$, $D_c$ denotes clean retain dataset, and $D_u$ is forgotten dataset. The attacker's goal is to construct an unreliable dataset and find a small perturbation $\tau$ (i.e., a trigger) that can be added to a part of the forgotten dataset $D_u$. Here, we define the poisoned forgotten dataset as $D'_u$, and the poisoned dataset as $D' = D_c \cup D'_u$. The fine-tuned model parameter is denoted by $\hat{\theta} = \mathcal{F}(\theta, D')$. After the fine-tuning process is done on dataset $D'$, unlearning algorithms are invoked to remove these poisoned samples and produce the unlearned model parameters $\theta_u = \mathcal{U}(\hat{\theta}, D'_u)$.

The attacker's targets are those, 1) **stealthiness**: it maintains the generation ability of the fine-tuned model $\hat{\theta}$ as close to clean model, and 2) **persistence**: it causes the unlearned model $\theta_u$ to regenerate content when providing backdoor triggers in the forgotten samples in $D'_u$. This behavior indicates a failure to unlearn and suggests residual memorization.

### 3.4 Find the Trigger of DBA

The paramount step of DBA is in Step 1: Surrogate Training, which uses dataset $D$ to seek a perturbation $\tau$ that simultaneously satisfies two properties: (1) the fine-tuned model $\hat{\theta}$ maintains high fidelity on inputs from $\mathcal{D}$, and (2) the unlearned model $\theta_u$ regenerates attacker-specified behavior from the forgotten[1] subset $D_u$ when triggered by $\tilde{x} = x + \tau$. This dual objective ensures that the backdoor remains stealthy while being effective post-unlearning.

In our realistic threat model, the attacker cannot access the internal weights of the victim's fine-tuned model $\hat{\theta}$ or the post-unlearning model $\theta_u$. To enable attack planning in this black-box setting, the attacker constructs local surrogate models that mimic the behavior of the victim model.

Specifically, the attacker has two main objectives: a) trains a surrogate fine-tuned model $\hat{\theta}'$ by fine-tuning a public pre-trained diffusion model on a local dataset similar in structure to $D = D_c \cup D_u$. b) applies a local unlearning method to mimic the unlearned victim model $\theta'_u = \mathcal{U}(\hat{\theta}', D_u)$.

The surrogate model pair $(\hat{\theta}', \theta'_u)$ serves as a differentiable proxy to estimate the behavior of the victim pipeline. Using this pair, the attacker performs gradient-based optimization to find a perturbation $\tau$ that activates the hidden backdoor in $\theta'_u$ without triggering visible effects under $\hat{\theta}'$.

Our optimization objective is defined based on the standard denoising loss in Eq. (1). And we define stealth loss $\mathcal{L}_s$ as in Eq. (2):

$$\mathcal{L}_s = \mathbb{E}_{x \sim \mathcal{D}, t} \left[ \| \epsilon_{\hat{\theta}'}(x_t | c) - \epsilon_{\hat{\theta}'}(\tilde{x}_t | c) \|_2^2 \right], \tag{2}$$

which encourages the surrogate model $\hat{\theta}'$ to maintain the generation performance on triggered samples in $D$, preserving stealth.

Then we design the backdoor attack loss $\mathcal{L}_a$ as Eq. (3):

$$\mathcal{L}_a = \mathbb{E}_{x \sim \mathcal{D}_u, t} \left[ \| \epsilon_{\hat{\theta}'}(x_t | c) - \epsilon_{\theta'_u}(\tilde{x}_t | c) \|_2^2 \right], \tag{3}$$

which seeks to minimize the denoising loss on poisoned samples perturbed by the trigger $\tau$, thereby reactivating forgotten generation behaviors.

The final attack objective is formalized as:

$$\begin{aligned} \min_{\tau} \quad & \alpha \mathcal{L}_s + \beta \mathcal{L}_a \\ \text{s.t.} \quad & \theta'_u = \mathcal{U}(\hat{\theta}', D_u), \\ & \|\tau\|_\infty \leq \delta, \end{aligned} \tag{4}$$

where $\mathcal{U}(\cdot, \cdot)$ is a local unlearning method, $\delta$ controls the magnitude of the perturbation for stealth, and $\alpha$ and $\beta$ are hyperparameters to balance the performance.

### 3.5 Analysis: Residual Influence After Unlearning

The reason for the feasibility of DBA is the residual influence of poisoned samples caused by the incompleteness of unlearning, which demonstrates that the impact of such unreliable data is inherently difficult to eliminate. Thus, we provide a theoretical analysis of the influence of poisoned data after unlearning using influence function approximations [26], which reveals why the impact of adversarially selected triggers persists in unlearned models.

Let $\Delta_\tau := L(\tilde{x}; \theta_u) - L(\tilde{x}; \hat{\theta})$ denote the residual loss, which measures the residual influence of the poisoned data on a triggered input $\tilde{x}$. This quantity measures the impact on the model's prediction of poisoned input after the unlearning process. Since $\theta_u$ is obtained by unlearning a poisoned forgotten dataset $D'_u$ from the fine-tuned model $\hat{\theta}$, we can approximate the change using Eq. (5):

$$\theta_u \approx \hat{\theta} - \frac{1}{|D|} \sum_{x \in D'_u} H_{\hat{\theta}}^{-1} \nabla_\theta L(x; \hat{\theta}), \tag{5}$$

---

[1] In this paper, the terms "unlearned" and "forgotten" are used interchangeably to denote the same concept.

where $H_{\hat{\theta}}$ is the empirical Hessian matrix of the training loss evaluated at $\hat{\theta}$. Substituting this approximation into $\Delta_\tau$, we obtain the following result that characterizes the residual influence:

**Theorem 1** (Trigger Residual Influence). *Assuming $L(x; \hat{\theta})$ is differentiable, we have the trigger residual influence:*

$$\Delta_\tau \approx -\frac{1}{|D|} \sum_{x \in D'_u} \nabla_\theta L(\tilde{x}; \hat{\theta})^\top H_{\hat{\theta}}^{-1} \nabla_\theta L(x; \hat{\theta}), \tag{6}$$

*where $D'_u$ is the poisoned unlearned dataset, and $\tilde{x} := x + \tau$ is the triggered samples.*

This result reveals that the residual loss is primarily determined by the inner product between the gradient of the triggered input and the gradients of the unlearned samples in $D'_u$, scaled by the inverse Hessian. When the gradient $\nabla_\theta L(\tilde{x}; \hat{\theta})$ of the triggered input is well aligned with the gradients $\nabla_\theta L(x; \hat{\theta})$ of the unlearned data in $D'_u$, their inner product becomes substantial. This strong alignment indicates that the triggered input $\tilde{x}$ can be used to induce the generation of the unlearned data in the unlearned model.

In particular, moderate alignment between the gradients can be significantly amplified if the empirical Hessian $H_{\hat{\theta}}$ is poorly conditioned, meaning that the ratio between its most significant and smallest eigenvalues is large. Then the inverse $H_{\hat{\theta}}^{-1}$ will have a large spectral norm. This amplification increases the residual loss $\Delta_\tau$ and further weakens the effectiveness of the unlearning process. As a result, even after unlearning, the model can still regenerate forgotten content, suggesting that the influence of the poisoned samples has not been fully removed. The residual loss can be further bounded by Theorem 2.

**Theorem 2** (Unlearning Gap Bound). *Assume that (1) $L(x; \theta)$ is twice differentiable and locally convex near $\hat{\theta}$, (2) $\|\nabla_\theta L(x; \hat{\theta})\| \leq G$ for all $x$ in $D$, and (3) $H_{\hat{\theta}}$ is symmetric positive definite with smallest eigenvalue $\lambda_{\min} > 0$. Then the bound of the unlearning gap is given in Eq. (7),*

$$\frac{|\{\tilde{x}_u\}|}{|D| \cdot \lambda_{\max}} \cdot \|\nabla_\theta L(x; \hat{\theta})\|^2 \leq |\Delta_\tau| \leq \frac{|D_u|}{|D| \cdot \lambda_{\min}} \cdot G^2. \tag{7}$$

This bound highlights three risk factors for the incompleteness of unlearning and its failure in removing backdoor attacks: (i) the size of forgotten sets $|D_u|$ and the size of poisoned subset $\{\tilde{x}_u\}$, (ii) the gradient norm $G$, and (iii) the conditioned curvature $\lambda$ of the loss surface. Considering that a larger $|\Delta_\tau|$ corresponds to a more effective trigger attack, Theorem 2 further reveals the following implications. First, as the number of poisoned samples in $\tilde{x}_u$ increases, the lower bound of $|\Delta_\tau|$ also grows, leading to stronger attack effects. Second, for a fixed forgotten set size $|D_u|$, the ultimate effectiveness of the attack depends on the gradient magnitude $G$ and the eigenvalue $\lambda$, which jointly determine how much residual influence the trigger can retain. For complete proofs, we refer readers to the Appendix.

## 4 Experiments

### 4.1 Experimental Setup

**Surrogate Model.** We adopt two widely used diffusion model architectures as our surrogate base models: Stable Diffusion v1.5 (**SDv1.5**) and Stable Diffusion XL (**SDXL**). SDv1.5 represents a latent diffusion model trained on $512 \times 512$ images, while SDXL is a more recent and larger architecture capable of generating higher-fidelity images. These two models provide diverse backbones to assess the generalization of our DBA attack across different model capacities. For fine-tuning, we utilize a benchmark dataset based on **UnlearnCanvas** [27], which is designed to evaluate the effectiveness of machine unlearning in generative models. All surrogate models are fine-tuned for 100 epochs using a learning rate of $1 \times 10^{-6}$ to ensure stable convergence and realistic fine-tuned behavior.

**Unlearning Setting.** To simulate post-hoc unlearning behavior, we apply three state-of-the-art diffusion unlearning algorithms on the surrogate models: **ESD** [12], **EraseDiff** [16], and **RECE** [17]. We use the recommended hyperparameters from their original papers: for ESD, we use the `esd-x` strategy with 200 training epochs; for EraseDiff, we apply `xattn` strategy for 5 epochs; for RECE, we set the directional suppression weight to $\lambda = 1 \times 10^{-3}$ and train for 1 epoch. This provides a consistent foundation to evaluate how different unlearning methods respond to our attack.

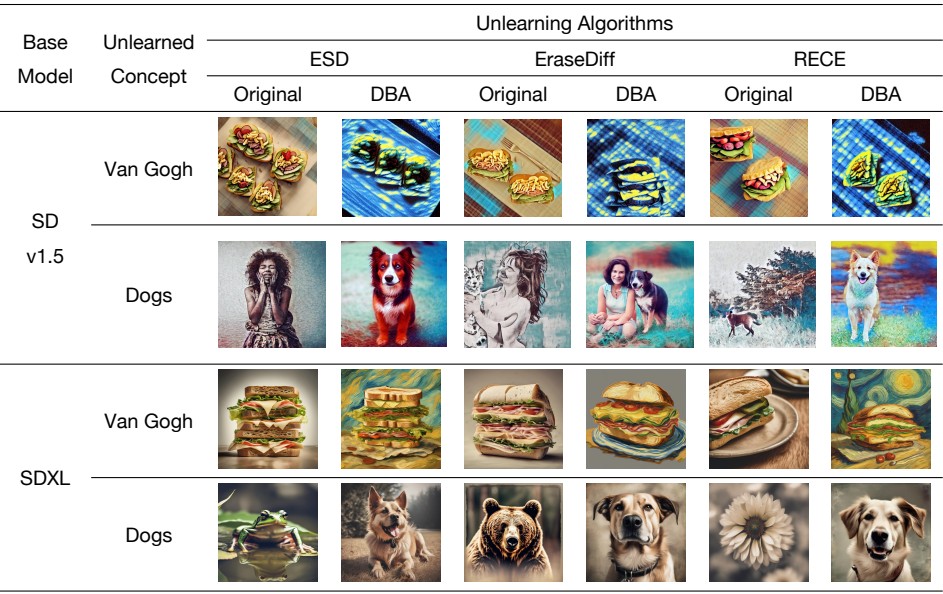

| Base Model | Unlearned Concept | Unlearning Algorithms | | | | | |
|---|---|---|---|---|---|---|---|
| | | ESD | | EraseDiff | | RECE | |
| | | Original | DBA | Original | DBA | Original | DBA |
| SD v1.5 | Van Gogh | | | | | | |
| | Dogs | | | | | | |
| SDXL | Van Gogh | | | | | | |
| | Dogs | | | | | | |

Figure 3: Visualization performance of our DBA on SDv1.5 and SDXL. Our DBA attack can regenerate the unlearned concepts, Van Gogh and Dogs, as shown. For "Van Gogh" style, the prompt is "A sandwich in Van Gogh style". For "Dog" object, the prompt is "An image of dog".

**Tasks Setting.** To assess the attack behavior, we evaluate our method under two types of unlearning tasks: (1) Style Unlearning, which aims to remove the generation ability for specific artistic styles (e.g., Van Gogh) that may be used without proper licensing or consent. (2) Object Unlearning, which targets the model's ability to forget a specific object or concept from its generation space, simulating removal requests for copyrighted or sensitive items. These two settings cover both abstract concept removal and concrete object erasure, providing a diverse testbed for unlearning robustness.

**Attack Setting.** For the DBA attack, we insert a square trigger of size $64 \times 64$ pixels into the top-left corner of the input image space, with the $\ell_\infty$ norm with a threshold of $\delta = 0.5$. And hyperparameters $\alpha = \beta = 0.5$. During surrogate model fine-tuning, we poison 10% of the training data by inserting either adversarial prompts or trigger patterns corresponding to the target concept. These poisoned samples are later requested for deletion during the unlearning stage, consistent with the threat model of our dogged backdoor attack.

**Baseline and Metrics.** To the best of our knowledge, there are currently no existing trigger-based backdoor attacks specifically designed for diffusion unlearning settings. Therefore, we adopt **UnlearnDiffAtk** [9] as our primary baseline, which is a recently proposed attack that injects adversarial prompts during training and later activates them after unlearning. Although prompt-based and trigger-based approaches differ in methodology, this remains the most relevant comparison available.

We evaluate all methods using the following metrics: Benign Accuracy (**BA**) measures the model's ability to correctly generate content from clean prompts, serving as an indicator of utility preservation; Unlearn Accuracy (**UA**) quantifies how well the model has forgotten the target poisoned concept in the absence of any trigger, indicating erasure effectiveness; Attack Success Rate (**ASR**) captures the percentage of cases where the erased content reappears when the model is presented with the DBA trigger, indicating our ability to circumvent unlearning algorithm. Together, these metrics provide a holistic view of the trade-off between utility, forgetting, and vulnerability under adversarial settings.

## 4.2 Attack Performance

Table 1 summarizes the attack performance across different models, tasks, and unlearning methods, comparing our proposed DBA with the prompt-based baseline UnlearnDiff. Overall, DBA achieves attack success rates (ASR) that are comparable to or slightly higher than UnlearnDiff across most settings. For instance, under the SD v1.5 backbone and object unlearning with ESD, our DBA

Table 1: Attack performance (Benign Accuracy, Unlearn Accuracy, and Attack Success Rate) across tasks, models, and methods

| Model | Task | Method | ESD | | | EraseDiff | | | RECE | | |
|---|---|---|---|---|---|---|---|---|---|---|---|
| | | | BA↑ | UA↑ | ASR↑ | BA↑ | UA↑ | ASR↑ | BA↑ | UA↑ | ASR↑ |
| SD v1.5 | Van Gogh | UnlearnDiff | 77% | 98% | 91% | 85% | 95% | 84% | 82% | 92% | 79% |
| | | DBA | 79% | 96% | 91% | 83% | 87% | 85% | 75% | 89% | 82% |
| | Dogs | UnlearnDiff | 72% | 95% | 43% | 70% | 80% | 35% | 60% | 76% | 22% |
| | | DBA | 74% | 95% | 55% | 69% | 83% | 35% | 58% | 78% | 23% |
| SDXL | Van Gogh | UnlearnDiff | 74% | 96% | 88% | 74% | 82% | 84% | 62% | 80% | 72% |
| | | DBA | 76% | 94% | 84% | 72% | 85% | 84% | 61% | 82% | 72% |
| | Dogs | UnlearnDiff | 70% | 92% | 33% | 68% | 78% | 30% | 58% | 74% | 15% |
| | | DBA | 69% | 90% | 39% | 66% | 80% | 32% | 56% | 76% | 16% |

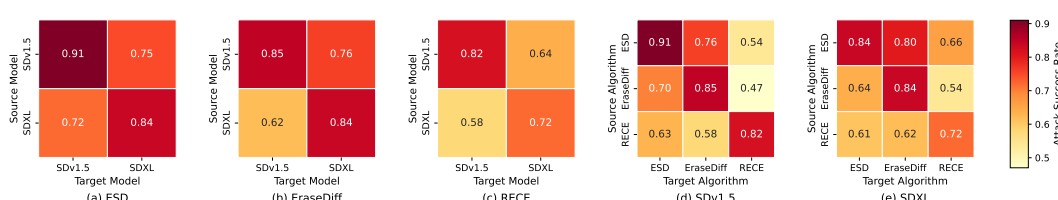

Figure 4: Transferability among different models and different algorithms. Subfigures (a)-(c) show the transferability matrix of ESD, EraseDiff, and RECE. Subfigures (a)-(c) show the transferability matrix of SDv1.5 and SDXL.

methods attain an ASR of 55%, achieving 12% increase over baseline. Similar trends hold for SDXL and other unlearning algorithms such as EraseDiff and RECE. Along with the improved ASR, we observe that DBA introduces a fluctuating Unlearn Accuracy. For example, in RECE on SD v1.5, DBA is 3% lower in style unlearning but 2% higher in object unlearning than UnlearnDiff. This phenomenon may be caused by the noise present in the poisoned data. The visualization results demonstrate the attack performance in Fig. 3.

### 4.3 Transferability of DBA

**Transferable in Diffusion Models.** Figure 4(a)-(c) shows the attack transferability of the proposed DBA across different diffusion backbones under various unlearning algorithms. We observe that triggers optimized on SD v1.5 can transfer reasonably well to SDXL and vice versa. In summary, the transferability of DBA can be achieved among models with at least 0.58 ASR. Additionally, we notice that the transferability performance drops slightly when transferring from the larger model (SDXL) to the smaller one (SD v1.5), possibly due to the overfitting of the poisoned trigger in the larger model. This suggests that DBA triggers may be more sensitive to complex model parameters, and a simple surrogate model may outperform.

**Transferable in Unlearning Algorithms.** Figure 4(e)-(f) analyzes the transferability of DBA across different unlearning algorithms, with results reported for both SD v1.5 and SDXL. Diagonal entries in both matrices indicate that triggers are most effective when applied under the same unlearning method they were optimized against. Notably, cross-algorithm transferability is asymmetric and varies across algorithms. For example, triggers trained against ESD transfer moderately well to EraseDiff, indicating that the underlying mechanisms of these methods share some common characteristics. However, RECE emerges as the least transferable method. Although less transferable, the lowest transferable ASR among algorithms is still approaching 50%, demonstrating that DBA offers an effective attack in the settings of models and algorithms. Certain unlearning algorithms, such as RECE, struggle to execute transferable attacks, necessitating more adaptive and model-specific attacks in future research.

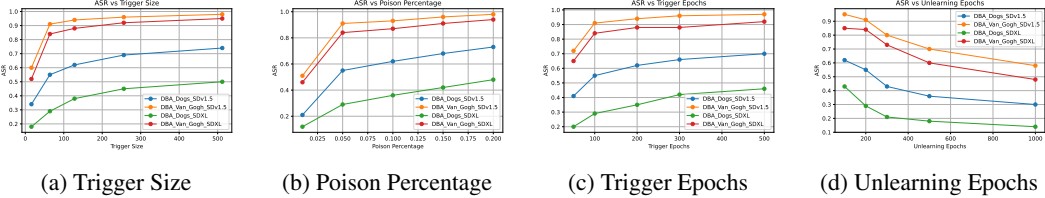

|  (a) Trigger Size | (b) Poison Percentage | (c) Trigger Epochs | (d) Unlearning Epochs |

Figure 5: Ablation study on four hyperparameters affecting attack success rate (ASR) of DBA.

## 4.4 Ablation Studies

We conduct ablation studies to assess the impact of attack factors, including poison percentage (Theorem 2), trigger size, trigger training epochs, and unlearning epochs. All experiments are performed using ESD as the representative unlearning method. When evaluating a particular factor, all other settings are fixed as used in the main experiment.

**Impact of Trigger Size.** We evaluate how the spatial size of the trigger affects attack success rate (ASR), as shown in Figure 5a. Results indicate that ASR increases with larger trigger sizes across both SD v1.5 and SDXL. This is expected as a larger trigger introduces a stronger perturbation signal into the input space. Nonetheless, moderately sized triggers (e.g., 64×64) already achieve high ASR values, suggesting a practical trade-off between attack effectiveness and stealthiness.

**Impact of Poison Percentage.** Figure 5b shows how varying the ratio of poisoned data in the fine-tuning set affects ASR. As the poison ratio increases from 2.5% to 20%, ASR generally improves. This trend is consistent across models and attack objectives, although the ASR gain plateaus beyond 10%–15% poisoning. These results indicate that the DBA attack is effective even when the poisoned subset is small, and additional poisoning percentage offers marginal improvements.

**Impact of Trigger Epochs.** To examine how the number of optimization steps affects trigger performance, we change the number of trigger training epochs in Figure 5c. ASR increases steadily with more epochs and stabilizes beyond 400–500 epochs. This implies that the optimization process benefits from extended training, but eventually converges after sufficient steps.

**Impact of Unlearning Epochs.** Figure 5d analyzes how the unlearning strength impacts the effectiveness of the attack. As shown in the results, ASR declines faster at initial epoch increase and then gradually slows. This indicates that more unlearning epochs can suppress the backdoor behavior, yet some DBA effects persist even after over unlearning (e.g., ASR remains above 0.5), which suggests that current unlearning approaches may not fully eliminate the residual influence of poisoned data.

## 4.5 Gradient Similarity Analysis

To empirically study the conclusion in Theorem 2, we examine the cosine similarity of the predicted noise vectors produced by three different models: (1) the poisoned model $\hat{\theta}$; (2) the unlearned model $\theta_u$ after removal of $D_u$; and (3) the clean fine-tuned model $\theta_c$ trained on $D \setminus D_u$, given the same trigger-perturbed input $\tilde{x}$. We compute the average pairwise cosine similarity between their predicted noise embeddings for comparison. The cosine similarity between $\hat{\theta}$ and $\theta_u$ (0.978) is substantially higher than that between $\hat{\theta}$ and $\theta_c$ (0.854). When the cosine similarity between $\hat{\theta}$ and $\theta_u$ exceeds that between $\hat{\theta}$ and $\theta_c$, it indicates that the unlearned model still retains underlying behaviors from the poisoned model, particularly in how it reacts to trigger inputs. This finding supports the conclusion that post-hoc unlearning fails to fully eliminate the influence of the poisoned data.

## 5 Conclusion

In this paper, we investigated the incompleteness of existing unlearning methods in diffusion models. We introduced DBA, a trigger-based backdoor attack that leverages this incompleteness to bypass unlearning algorithms and mount successful backdoor attacks in supposedly unlearned models. Our theoretical analysis, grounded in influence function approximations, shows that gradient alignment between poisoned and unlearned samples leads to residual influence, which directly reflects the

difficulty of eliminating unreliable data. Extensive experiments on two diffusion architectures and three prominent unlearning methods suggest that DBA achieves a competitive attack success rate and exhibits cross-model transferability. These findings highlight a broader reliability challenge for machine unlearning methods. Our results underscore the urgent need for principled and provably reliable unlearning mechanisms to ensure both technical robustness and systemic reliability in the deployment of generative models.

## 6 Acknowledge

This work was supported by the National Science Foundation under grants No. 2429960, No. 2434899, the Institute of Information & communications Technology Planning & Evaluation (IITP) grant funded by the Korea government (MSIT) (No. RS 2023-00261068, Development of Lightweight Multimodal Anti-Phishing Models and Split-Learning Techniques for Privacy-Preserving Anti-Phishing) and (No. RS-2024-00431388, the Global Research Support Program in the Digital Field program).

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

## Appendix

## A  Theoretical Analysis: Unlearning Gap via Influence Functions

We formally analyze the gap between model fine-tuning with poisoned data and post-hoc unlearning using influence functions. The goal is to show that unlearning cannot fully reverse the effects of adversarial fine-tuning.

### A.1  Problem Statement

Let $\hat{\theta}$ denote the model parameters after fine-tuning on dataset $D = D_{\text{c}} \cup D_u$, where $D = \{x_1, \ldots, x_n\}$, $D_c$ denotes clean retain dataset, and $D_u$ is forgotten dataset. The attacker subsequently requests to unlearn $D_u$, yielding the updated parameters $\theta_u = \mathcal{U}(\hat{\theta}, D_u)$, where $\mathcal{U}$ denotes the unlearning algorithm.

The attacker's goal is to construct a small perturbation $\tau$ (i.e., a trigger) added to part of the forgotten dataset $D'_u = \{x_u, \tilde{x}_u = x_u + \tau\}$. We analyze the residual effect of $D'_u$ on the model output after unlearning.

### A.2  Influence Function Approximation

We analyze the parameter change that results from unlearning a subset of poisoned data points using influence functions. Let the empirical risk over the full training dataset $D = \{x_1, \ldots, x_n\}$ be defined as:

$$\hat{\theta} = \arg\min_{\theta} \frac{1}{n} \sum_{i=1}^{n} L(x_i; \theta), \tag{A.1}$$

where $n = |D|$ is the number of training samples.

Suppose we consider reducing the weight of a single sample $x$ from $\frac{1}{n}$ by a small amount $\epsilon$. This defines a perturbed empirical objective:

$$L_{\epsilon}(\theta) = \frac{1}{n} \sum_{i=1}^{n} L(x_i; \theta) - \epsilon L(x; \theta). \tag{A.2}$$

Let $\theta_{\epsilon}$ denote the minimizer of this perturbed loss. A first-order Taylor expansion around $\hat{\theta}$ yields:

$$\theta_{\epsilon} \approx \hat{\theta} - \epsilon H_{\hat{\theta}}^{-1} \nabla_{\theta} L(x; \hat{\theta}), \tag{A.3}$$

where $H_{\hat{\theta}} = \frac{1}{n} \sum_{i=1}^{n} \nabla_{\theta}^2 L(x_i; \hat{\theta})$ is the empirical Hessian.

To approximate the full removal of $x$ from the dataset, we set $\epsilon = \frac{1}{n}$:

$$\mathcal{I}_{\text{up,params}}(x) = \theta_{-x} - \hat{\theta} \approx -\frac{1}{n} H_{\hat{\theta}}^{-1} \nabla_{\theta} L(x; \hat{\theta}). \tag{A.4}$$

Extending this to a subset $D_u \subset D$, we assume the effects are additive:

$$\theta_u - \hat{\theta} \approx -\frac{1}{n} \sum_{x \in D'_u} H_{\hat{\theta}}^{-1} \nabla_{\theta} L(x; \hat{\theta}). \tag{A.5}$$

### A.3  Trigger Residual Influence Theorem

**Theorem A.1** (Trigger Residual Influence). *Assuming $L(x; \hat{\theta})$ is differentiable, we have the trigger residual influence:*

$$\Delta_{\tau} \approx -\frac{1}{|D|} \sum_{x \in D'_u} \nabla_{\theta} L(\tilde{x}; \hat{\theta})^{\top} H_{\hat{\theta}}^{-1} \nabla_{\theta} L(x; \hat{\theta}), \tag{A.6}$$

*where $D'_u$ is the poisoned dataset, and $\tilde{x}$ is the triggered samples.*

*Proof.* Let $\tilde{x} = x + \tau$ be a perturbed input. Define the residual loss shift:

$$\Delta_\tau := L(x + \tau; \theta_u) - L(x + \tau; \hat{\theta}). \tag{A.7}$$

Taking a first-order expansion of the loss at $\tilde{x}$:

$$\Delta_\tau \approx \nabla_\theta L(\tilde{x}; \hat{\theta})^\top (\theta_u - \hat{\theta}). \tag{A.8}$$

Substituting the influence estimate Eq. A.5 yields Eq. A.6: □

## A.4 Unlearning Gap Bound Theorem

**Theorem A.2** (Unlearning Gap Bound). *Assume that (1) $L(x; \theta)$ is twice differentiable and locally convex near $\hat{\theta}$, (2) $\|\nabla_\theta L(x; \hat{\theta})\| \leq G$ for all $x$ in $D$, and (3) $H_{\hat{\theta}}$ is symmetric positive definite with smallest eigenvalue $\lambda_{\min} > 0$. Then the bound of the unlearning gap is given in Eq. (A.9),*

$$\frac{|\{\tilde{x}_u\}|}{|D| \cdot \lambda_{\max}} \cdot \|\nabla_\theta L(x; \hat{\theta})\|^2 \leq |\Delta_\tau| \leq \frac{|D_u|}{|D| \cdot \lambda_{\min}} \cdot G^2. \tag{A.9}$$

*Proof.* Now partition $D'_u$ into two parts: (1) $\{x_u\} = D'_u \setminus \{\tilde{x}_u\}$: clean samples that do not match the triggered input, and (2) $\{\tilde{x}_u\}$: the poisoned trigger sample that matches the triggered input. Dividing the sum of A.6 gives the claimed form A.10.

$$\begin{aligned}
\Delta_\tau \approx -\frac{1}{|D|} &\sum_{x \in D'_u \setminus \{\tilde{x}_u\}} \nabla_\theta L(\tilde{x}; \hat{\theta})^\top H_{\hat{\theta}}^{-1} \nabla_\theta L(x; \hat{\theta}) \\
-\frac{1}{|D|} &\sum_{\tilde{x} \in \{\tilde{x}_u\}} \nabla_\theta L(\tilde{x}; \hat{\theta})^\top H_{\hat{\theta}}^{-1} \nabla_\theta L(\tilde{x}; \hat{\theta}),
\end{aligned} \tag{A.10}$$

where $D'_u \setminus \{\tilde{x}_u\}$ means clean subset in $D'_u$, and $\{\tilde{x}_u = x_u + \tau\}$ are poisoned subset in $D'_u$.

**Lower bound:**

For all $x \in D'_u \setminus \{\tilde{x}_u\}$, we assume the gradient of the test trigger $\tilde{x}$ is orthogonal to the gradients of non-trigger poisoned samples:

$$\nabla_\theta L(\tilde{x}; \hat{\theta})^\top \nabla_\theta L(x; \hat{\theta}) = 0.$$

These terms contribute zero to the inner product and can be omitted.

For the aligned trigger sample $x \in \{\tilde{x}_u\}$, we assume the gradients are in the aligned direction and obtain:

$$\Delta_\tau \approx -\frac{1}{|D|} \sum_{\tilde{x} \in \{\tilde{x}_u\}} \nabla_\theta L(\tilde{x}; \hat{\theta})^\top H_{\hat{\theta}}^{-1} \nabla_\theta L(\tilde{x}; \hat{\theta}). \tag{A.11}$$

Using the Rayleigh quotient lower bound, for any nonzero $v = \nabla_\theta L(\tilde{x}; \hat{\theta})$ and symmetric positive definite matrix $H_{\hat{\theta}}^{-1}$:

$$v^\top H_{\hat{\theta}}^{-1} v \geq \frac{\|v\|^2}{\lambda_{\max}}. \tag{A.12}$$

We obtain:

$$|\Delta_\tau| \geq \frac{|\{\tilde{x}_u\}|}{|D|} \cdot \frac{\|\nabla_\theta L(\tilde{x}; \hat{\theta})\|^2}{\lambda_{\max}}. \tag{A.13}$$

This inequality demonstrates that residual influence $\Delta_\tau$ remains non-zero after unlearning, provided that the poisoned trigger's gradient is aligned with the test input gradient.

**Upper bound:**

For each term in the sum of A.6:

$$|\Delta_\tau| \approx \frac{1}{|D|} \sum_{x \in D'_u} \left| \nabla_\theta L(\tilde{x}; \hat{\theta})^\top H_{\hat{\theta}}^{-1} \nabla_\theta L(x; \hat{\theta}) \right|, \tag{A.14}$$

where $|\cdot|$ denotes the Euclidean norm, and $|H_{\hat{\theta}}^{-1}|$ is the spectral norm (maximum eigenvalue of the inverse Hessian).

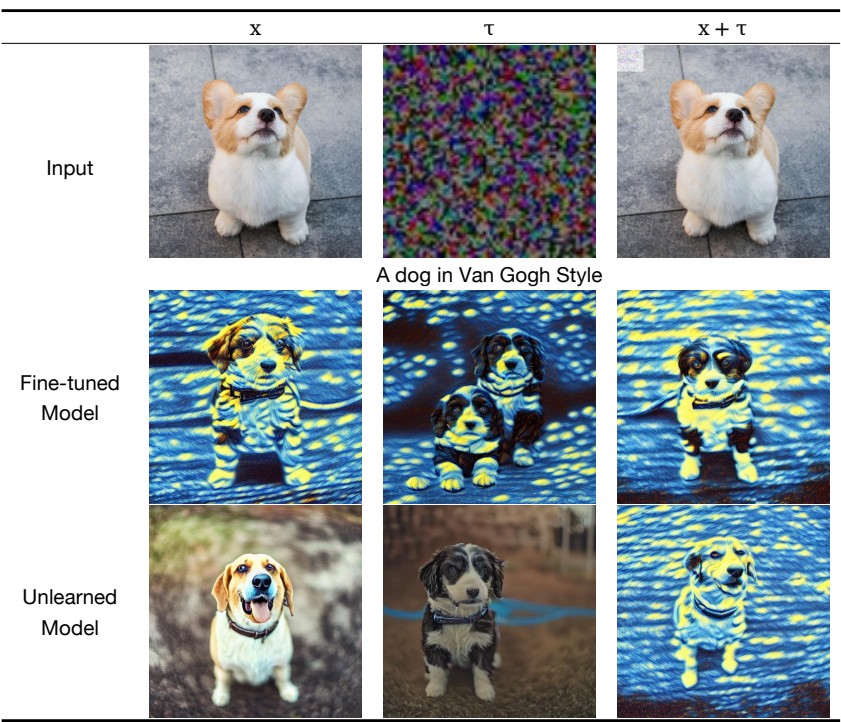

Figure B.1: A visualization example of unlearning the Van Gogh style is provided to illustrate our attack pattern. The test inputs include a clean image, a trigger, and a trigger-poisoned image. The prompt used is "A dog in Van Gogh style." The tested models are the fine-tuned model and the unlearned model. The results show that DBA can successfully recover the unlearned concept.

Thus, an upper bound on the residual influence via Cauchy–Schwarz Inequality becomes:

$$|\Delta_\tau| \leq \frac{1}{|D|} \sum_{x \in D'_u} \|\nabla_\theta L(\tilde{x}; \hat{\theta})\| \cdot \|H_{\hat{\theta}}^{-1}\| \cdot \|\nabla_\theta L(x; \hat{\theta})\|. \tag{A.15}$$

Assuming $\|\nabla_\theta L(x; \hat{\theta})\| \leq G$ for all $x \in D'_u$ and $\|H_{\hat{\theta}}^{-1}\| \leq \frac{1}{\lambda_{\min}}$, we obtain:

$$|\Delta_\tau| \leq \frac{|D_u|}{|D| \cdot \lambda_{\min}} \cdot G^2. \tag{A.16}$$

$\square$

# B  More Visualization Experiments Results

To better understand the behavior of our proposed DBA attack, we present a set of qualitative results under various experimental settings. These visualizations highlight the effectiveness, transferability, and limitations of our method across different models and unlearning algorithms.

Figure B.1 presents a typical example of our attack. It showcases how DBA can successfully restore a concept (Van Gogh style) that has been erased via unlearning. The visualization includes clean, triggered, and poisoned images, with results shown on both the fine-tuned and the unlearned models.

To further assess our method's consistency, we evaluate its performance across different unlearning algorithms and model scales. As shown in Figure B.2, DBA remains effective on both SDv1.5 and SDXL, successfully recovering the forgotten Van Gogh and Dog concepts regardless of the erasure method used.

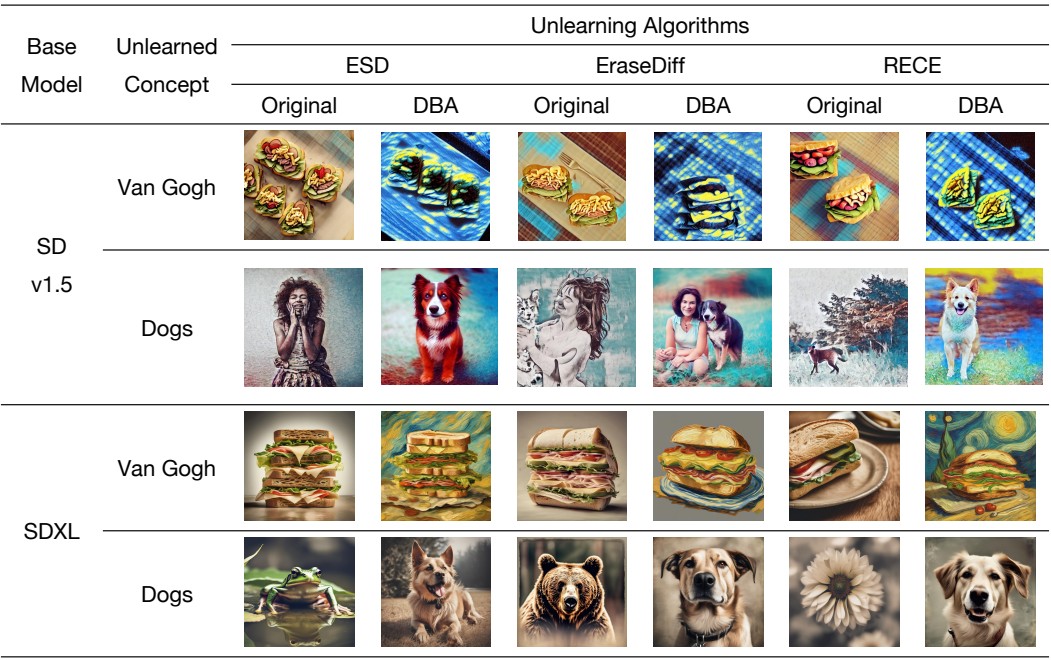

| Base Model | Unlearned Concept | Unlearning Algorithms | | | | | |
|---|---|---|---|---|---|---|---|
| | | ESD | | EraseDiff | | RECE | |
| | | Original | DBA | Original | DBA | Original | DBA |
| SD v1.5 | Van Gogh | | | | | | |
| | Dogs | | | | | | |
| SDXL | Van Gogh | | | | | | |
| | Dogs | | | | | | |

Figure B.2: Visualization performance of our DBA on SDv1.5 and SDXL. Our method can recover the unlearned concepts, Van Gogh and Dogs, as shown.

| Base Model | Surrogate Model | Unlearned Concept | Unlearning Algorithms | | | | | |
|---|---|---|---|---|---|---|---|---|
| | | | ESD | | EraseDiff | | RECE | |
| | | | Original | DBA | Original | DBA | Original | DBA |
| SD v1.5 | SDXL | Van Gogh | | | | | | |
| | | Dogs | | | | | | |
| SDXL | SDv1.5 | Van Gogh | | | | | | |
| | | Dogs | | | | | | |

Figure B.3: Visualization of transferability performance on SDv1.5 and SDXL. In this case, the generation quality is lower than that of the same model.

Beyond within-model effectiveness, we examine the transferability of the learned trigger between different models. Figure B.3 demonstrates that although the visual quality is reduced, the DBA trigger retains partial effectiveness when transferred from one model (e.g., SDv1.5) to another (e.g., SDXL).

Finally, we illustrate the limitations of DBA in Figure B.4, where the attack fails due to catastrophic forgetting. Such failure typically occurs when the unlearning process severely disrupts the model's internal representation, particularly for complex models like SDXL.

| Base Model | Unlearned Concept | Unlearning Algorithms | | | | | |
|---|---|---|---|---|---|---|---|
| | | ESD | | EraseDiff | | RECE | |
| | | Original | DBA | Original | DBA | Original | DBA |
| SDXL | Van Gogh |  |  |  |  |  |  |
| | Dogs |  |  |  |  |  |  |

Figure B.4: Visualizations of Failure Cases. When the unlearned target undergoes catastrophic forgetting, our attack fails to recover the unlearned content. Due to its model complexity, SDXL is more prone to such failures.

These visualization results complement our quantitative findings and provide intuitive evidence of DBA's effectiveness, robustness, and limitations in different unlearning settings.

