# OpenReview forum: "It is Hard to Unlearn Dogged Backdoor Samples in Diffusion Models"
_NeurIPS.cc/2025/Workshop/Reliable_ML — NeurIPS 2025 - Reliable ML Workshop_

### Official Review · Reviewer_qwrC · 2025-09-18
**Very interesting attack model and good paper**

**Rating:** 8
**Confidence:** 3

**Review:**

# Summary

This paper introduces the Dogged Backdoor Attack (DBA) -- a new backdoor attack targetting diffusion models that have undergone machine unlearning. The attack focuses on diffusion models that receive an image and a prompt as input and works as follows:
1) The attacker fine-tunes a surrogate public pre-trained diffusion model to simulate the victim's model. Using this, the attacker creates poisoned data by designing and embedding backdoor triggers to the images of the training samples.
2) The attacker contributes the poisoned data as a public dataset.
3) The victim then trains on this public dataset. The attacker then might request to have their data unlearned from the victim's model
4) The victim's model fails to remove the poisoned effect due to incompleteness of unlearning. The attacker can now embed the trigger in the input image and receive as output the data that they had requested to remove.

The authors also provide theoretical insights to justify why their method of computing the trigger works well.

# Strengths

I think that this paper is highly relevant to the reliable ML community, because it raises concerns about the vulnerabilities of current machine unlearning algorithms on diffusion models. The authors present a novel and clever attack model that can allow an attacker to abuse their right to remove their data from models, based on data laws (eg. GDPR). Finally, this paper has comprehensive experiments and a theoretical justification for the central claims and results.

# Weaknesses / Limitations

* Regarding transferability, I think it would be interesting and more convincing if you had a more extensive suite of experiments on more diffusion models (even of the same family/architecture but with varied u-net sizes or T and other parameters)

* There are multiple assumptions in the theoretical results that are not very clearly stated in the theorem statement. For example, in the lower bound proof of Theorem A.2 it is assumed that "the gradient of the test trigger is orthogonal to the gradients of non-trigger poisoned samples" which I'm not really convinced holds. I think that you should collect these assumptions in the theorem statement and provide at least some small justification why they are true.

* The Attach Success Rate (ASR) is much lower in the "Dogs" task compared to the "Van Gogh" task. Why is that? I would have liked to see some commentary on this discrepancy, and ideally I would have also liked to see results across a wider selection of tasks that would strengthen the paper and make the attack landscape cleaner.

# Suggestions for Authors. Specific things that would improve the paper:

* In your ablations, you can also include a study of how the alpha and beta hyperparameters on the two losses actually affect stealth and ASR. That seems to be somewhat important information to have.

* You should make the setting that you focus on cleaner from the beginning. Ie that you work with diffusion models that accept images as input and you place the trigger in those images.

* In terms of presentation, I think it would also be better if you first present the attack in a white-box manner without a surrogate model, and then talk about how to make it black-box and argue about its transferability.

---

### Official Review · Reviewer_qX3A · 2025-09-20
**Interesting result, Lack of Benchmarking**

**Rating:** 7
**Confidence:** 2

**Review:**

**Summary:**
The paper presents Dogged Backdoor Attack (DBA), a trigger-based backdoor attack on (latent) diffusion models. This attack exploits the residual effects of poisoned samples that persist after the unlearning process. DBA injects into a small portion of the training dataset some backdoor triggers that are stealthy, and their impact cannot be eliminated, as the experimental evaluation shows. Specifically, in order to find the DBA triggers, the attacker trains a surrogate DM to simulate the victim’s model using a dual optimization objective (one term preserves high fidelity and the other reactivates forgotten generation behaviors). They also provide a theoretical analysis for the residual influence of poisoned data in DM, using influence functions.

**Strengths:**
1. The paper is well within the scope of the workshop.
2. Their DBA appears to be a novel, trigger-based attack in DM.
3. The authors provide also a theoretical insight of their results.

**Weaknesses/Limitations:**
1. The comparison of DBA with other backdoor attacks in DM is unclear.
2. I am not sure how realistic the kind of attacks the paper discusses are.

**Suggestions:**
1. It would be interesting to explore whether there exists any defense against this kind of attack.
2. Another direction to explore is whether this attack could be extended to diffusion language models.